# Recovery of Water Homeostasis in Adenine-Induced Kidney Disease Is Mediated by Increased AQP2 Membrane Targeting

**DOI:** 10.3390/ijms25063447

**Published:** 2024-03-19

**Authors:** Jasmine C. L. Atay, Søren H. Elsborg, Johan Palmfeldt, Lene N. Nejsum, Rikke Nørregaard

**Affiliations:** 1Department of Clinical Medicine, Aarhus University, 8200 Aarhus N, Denmark; jcla@clin.au.dk (J.C.L.A.); sohe@clin.au.dk (S.H.E.); nejsum@clin.au.dk (L.N.N.); 2Research Unit for Molecular Medicine, Department of Clinical Medicine, Aarhus University, 8200 Aarhus N, Denmark; johan.palmfeldt@clin.au.dk; 3Department of Renal Medicine, Aarhus University Hospital, 8200 Aarhus N, Denmark

**Keywords:** aquaporin, chronic kidney disease, kidney fibrosis, regeneration

## Abstract

Chronic kidney disease (CKD) represents a major public health burden with increasing prevalence. Current therapies focus on delaying CKD progression, underscoring the need for innovative treatments. This necessitates animal models that accurately reflect human kidney pathologies, particularly for studying potential reversibility and regenerative mechanisms, which are often hindered by the progressive and irreversible nature of most CKD models. In this study, CKD was induced in mice using a 0.2% adenine-enriched diet for 4 weeks, followed by a recovery period of 1 or 2 weeks. The aim was to characterize the impact of adenine feeding on kidney function and injury as well as water and salt homeostasis throughout disease progression and recovery. The adenine diet induced CKD is characterized by impaired renal function, tubular injury, inflammation, and fibrosis. A significant decrease in urine osmolality, coupled with diminished aquaporin-2 (AQP2) expression and membrane targeting, was observed after adenine treatment. Intriguingly, these parameters exhibited a substantial increase after a two-week recovery period. Despite these functional improvements, only partial reversal of inflammation, tubular damage, and fibrosis were observed after the recovery period, indicating that the inclusion of the molecular and structural parameters is needed for a more complete monitoring of kidney status.

## 1. Introduction

Chronic kidney disease (CKD) represents a significant health concern affecting approximately 10% of the global population, and its prevalence is steadily increasing [1]. CKD is characterized by the development of kidney fibrosis and progressive loss of kidney function and will ultimately progress to end-stage renal disease (ESRD). Current therapies primarily serve to delay the progression of CKD to ESRD, highlighting a need for treatment strategies that aim at reversing the condition [2]. To achieve this, it is important to gain a comprehensive understanding of the mechanistic processes that drive CKD pathogenesis.

Animal models of CKD that mirror the complexities of the human condition play an important role in understanding the pathogenesis of the disease. However, most CKD animal models are induced by irreversible injury, making it challenging to study the possible regenerative mechanisms of kidney injury [3]. A deeper understanding of reversibility can facilitate the development of novel therapeutic targets that could potentially promote kidney function recovery in clinical settings. The adenine-induced CKD model, first described by Yokozawa et al. [4], is characterized by a progressive manifestation of kidney damage. This progression is demonstrated by increasing levels of plasma creatinine and blood urea nitrogen (BUN), coupled with a concurrent increase in tubular damage [5,6]. Moreover, this model is marked by a gradual development of inflammation and interstitial fibrosis, highlighting its utility in simulating and studying the multifaceted aspects of CKD. In this model, kidney pathology is thought to occur due to the formation of 2,8-dihydroxyadenine (2,8-DHA), an adenine metabolite that forms crystals within the tubular lumen, affecting the tubule cells [4,7,8]. Interestingly, adenine also acts as a signaling molecule through the adenine receptor Ade2R and causes fluid and salt wasting after 7 days of adenine feeding, which has been suggested as an early mechanism of adenine-induced kidney injury [9]. Hence, the impact of adenine-induced CKD over an extended duration on renal water and salt homeostasis remains unclear. Additionally, the question arises whether reversibility of adenine-induced CKD affects water and salt balance. 

We hypothesized that adenine feeding for 4 weeks induces a severe state of CKD and that a subsequent recovery period of 2 weeks will result in the improvement of kidney function, with concurrent improvement in water and salt homeostasis. We anticipated that this timeframe would be sufficient to observe significant changes in kidney function and pathology. To study this, we employed a reversible adenine-induced CKD model, adapting it to investigate the dynamic effects on kidney function and injury as well as water and salt homeostasis, with a focus on key water and salt transporters throughout the phases of disease progression and recovery. 

## 2. Results

### 2.1. Recovery of Adenine-Induced CKD Improves Kidney Function

To induce severe CKD, mice were subjected to an adenine-enriched diet for 4 weeks, followed by a switch to the control diet for another 1 or 2 weeks to examine the potential recovery of the kidneys (Figure 1A). Each group contained at least 7 mice. Adenine-fed mice exhibited weight loss already after day 2, which stabilized around day 14 (Figure 1B). Following the switch to the control diet, body weight (BW) started to increase, and after 7 days, the BW of the 4+2-week group was comparable to normal control mice. Mice were placed in metabolic cages to measure individual food and water intake as well as urine output. During the adenine-feeding phase, food intake was significantly reduced in the 4-week group and started to normalize during the recovery (Table 1). Water intake was significantly increased in the 4+2-week group when compared to control and the 4-week group. Mice fed with adenine-enriched diet for 4 weeks showed increased urinary excretion of adenine and the 2,8-DHA metabolite, which was decreased to similar levels as in control mice after recovery (Table 1). The levels of both metabolites were below detection level in the plasma. Kidney function, as measured by plasma creatinine and BUN, was impaired following 4 weeks of adenine feeding, and both functional parameters improved during the recovery phase (Figure 1C,D). In contrast to the significant improvement in functional parameters, the kidneys persistently decreased in weight and exhibited shrinkage and less healthy tubule mass on the macroscopic level during the recovery phase (Figure 1E,F). 

### 2.2. Effect of Adenine and the Subsequent Recovery Period on Water and Salt Homeostasis

To further evaluate whether recovery of adenine-induced CKD affects the kidney’s ability to regulate the urinary concentration capacity, we investigated the regulation of water and salt homeostasis. To measure changes in urine output, urine was collected before the start and at the end of the experiment. Following adenine feeding, urine output was increased by 400% relative to baseline in the 4-week group (*p* = 0.09) and by 240% in the 4+1-week group (*p* < 0.005) (Figure 2A). However, we observed no significant changes between baseline and terminal urine output in the 4-week group, although data showed noticeable variation. No change in urine output between baseline and terminal was observed in the 4+2-week group. Urine osmolality was significantly lower in all groups initially fed the adenine diet and showed a tendency to normalize during the recovery period (Figure 2B,C). However, following 2 weeks of recovery, urine osmolality was significantly higher compared with the 4-week adenine group (Figure 2C). Plasma osmolality and sodium increased significantly in the 4+2-week group compared to the control and 4-week group, respectively (Table 2). No change was observed in plasma potassium. Urinary sodium and potassium excretion corrected for food intake did not significantly differ from controls in any groups, although there was a trend towards increased sodium excretion in the 4-week group compared to control (Table 2). No proteinuria was detected across the groups (Table 2). Together, these findings suggest that the adenine diet leads to imbalances in the regulation of water and salt in the kidneys which, to some extent, can be improved during the recovery phase.

### 2.3. Recovery of Adenine-Induced CKD Is Associated with Increased Apical Targeting of AQP2 in Inner Medullary Collecting Ducts

To understand the molecular mechanisms for altered water and salt balance as well as urine osmolality, we measured several key water and salt transporters in the cortex and medulla of the kidney on both the mRNA and protein levels. AQP2 localizes to the apical plasma membrane and to the subapical intracellular vesicles in collecting duct principal cells and plays a central role in regulating urine concentration. Upon activation, AQP2 is shuttled from these subapical membranes into the plasma membrane; thus, AQP2 function can be studied by measuring the basal-to-luminal distribution of AQP2 [10]. Our data showed that adenine feeding was associated with a significant downregulation of AQP2 on both mRNA and protein levels in the inner medulla. During the recovery phase, AQP2 mRNA expression increased significantly compared with the 4-week group (Figure 2D). Additionally, a line scan analyzing the subcellular localization of AQP2 revealed increased apical plasma membrane targeting of AQP2 in the inner medullary collecting ducts during the recovery phase (Figure 2E,G). 

Due to the signaling effects of adenine in the rodent kidney [9], we explored the gene expression of the adenine receptor Ade2R and observed an initial increase after 4 weeks of adenine feeding. Although the expression showed a gradual return toward control levels during recovery, there was a tendency for it to remain elevated (Figure 2F). Together, these analyses revealed a disturbed regulation of water homeostasis in adenine-fed mice. This disturbance can be explained by decreased expression levels and plasma membrane targeting of AQP2, as well as the signaling effects of adenine via the increased expression of the Ade2R.

Besides AQP2, other AQPs, also expressed in the collecting duct, are responsible for the reabsorption of water, namely AQP3 and AQP4 both localize to the basolateral membrane of the principal cells of the collecting ducts [11]. We observed a significant downregulation of AQP3 and AQP4 mRNA expression in the inner medulla following adenine feeding (Figure 3A,B), which was also confirmed by immunohistochemistry staining, where fewer tubules in the collecting duct stained positive for AQP3 and AQP4 (Figure 3C,D). While AQP3 expression returned to control levels during the recovery phase, the expression of AQP4 remained low with a tendency to normalize after the 2-week recovery period. This suggests that AQP2-4 potentially plays a role in the observed increase in urine-concentrating capacity during recovery of adenine-induced CKD.

### 2.4. Effect of Adenine and the Subsequent Recovery Period on the Expression of AQP1 and Na^+^-Dependent Transporters (NHE3 and NKCC2) in Adenine-Induced CKD

AQP1 plays an important role in the water reabsorption in the proximal tubules [11]. However, loss of AQP1 has also been associated with increased tubular injury and inflammation [12]. Our data demonstrated that mRNA expression of AQP1 was not significantly changed throughout the study period (Figure 4A). However, immunofluorescence staining revealed a reduction in AQP1-positive proximal tubules within the cortical region at 4 weeks, in line with our observation that adenine feeding is associated with less healthy proximal tubule mass. Recovery did not affect the AQP1 staining pattern (Figure 4B,C). Another proximal tubular transporter, the sodium hydrogen exchanger 3 (*Slc9a3*, NHE3), was also examined. NHE3 mRNA expression was reduced in the 4-week and 4+1-week groups compared to controls, but no significant change was observed between the 4+2-week group and controls (Figure 4D).

The sodium–potassium–chloride cotransporter 2 (*Slc12a1*, NKCC2) in the medullary thick ascending limb (mTAL) is crucial for urine dilution in TAL and the formation of a hypertonic medullary interstitium and is essential for water reabsorption in the collecting duct system [13]. We observed a significant increase in NKCC2 mRNA expression following 1 week of recovery compared to the controls and 4-week group (Figure 4E). However, protein levels of NKCC2 significantly decreased during adenine feeding and remained decreased throughout recovery (Figure 4F,G). Altogether, these data indicate that recovery did not considerably affect the levels of the proximal transporters (AQP1 and NHE3) or NKCC2. 

### 2.5. Effect of Adenine and the Subsequent Recovery Period on Inflammation and Tubular Injury in Adenine-Induced CKD 

Two of the hallmarks in the progression of kidney disease are tubular and interstitial inflammation and fibrosis [14]. To investigate whether kidney inflammation and tubular injury are affected by a recovery phase in adenine-induced CKD, we examined the expression of inflammatory and tubular injury markers in the kidney cortex. mRNA expression levels of proinflammatory markers interleukin 6 (*Il6*), interleukin 1 beta (*Il1b*), tumor necrosis factor (*Tnf*), and C-C motif chemokine ligand 2 (*Ccl2*) were all significantly increased following adenine feeding (Figure 5A–D). During the recovery phase, the expression of *Il6* and *Il1b* were significantly decreased following 1 and 2 weeks of recovery compared with the 4-week group. *Tnf* and *Ccl2* expression also decreased in the 4+2 group, but it did not reach significance. The tubular injury marker, kidney injury molecule-1 (KIM-1, also known as hepatitis A virus cellular receptor 1 (*Havcr1*)) mRNA expression, increased in adenine-fed mice, and its expression significantly decreased during the recovery period (Figure 5E). This was also confirmed by immunofluorescence staining showing an increased percentage of KIM-1-positive areas in the cortical region in the adenine-fed groups compared to the controls, and during recovery, the KIM-1 staining pattern was markedly reduced (Figure 5G). These data indicate that adenine feeding is associated with increased inflammation and kidney tubule injury and these changes can be partially reversed during the recovery of adenine-induced CKD. 

### 2.6. Effect of Adenine and the Subsequent Recovery Period on Fibrosis in Adenine-Induced CKD 

Next, we explored the impact of adenine exposure and the following recovery period on different fibrosis markers. During the adenine-enriched diet, collagen 1a1 (*Col1a1*), fibronectin (*Fn1*), α-smooth muscle actin (*Acta2,* α-SMA), and transforming growth factor-β (*Tgfb1*) mRNA expression progressively increased in cortical tissue and remained elevated throughout the recovery phase (Figure 6A–D). These data were confirmed by Western blotting analyses showing that both fibronectin (FN) and α-SMA protein levels were elevated in both the 4-week and 4+1-week groups compared to controls. After a 2-week recovery period, there was a significant reduction in FN protein levels compared to the 4+1-week group and a significant decrease in α-SMA protein levels compared to the 4-week group (Figure 6E,F). Sirius Red staining revealed predominant collagen deposition during the adenine feeding with persistent fibrosis at both time points of the recovery phase (Figure 6G,H). Immunohistochemical staining of FN showed strong and increased labeling in both cortex and inner medulla, which partly decreased during the 2 weeks of recovery (Figure 6I,J). Double immunofluorescence staining of α-SMA and platelet-derived growth factor receptor-β (PDGFR-β) showed increased labeling in all adenine-fed groups, and this increase was most likely due to activation and expansion of interstitial mesenchymal cells (Figure 6K). Additionally, we observed an accumulation of periglomerular α-SMA+ cells in the 4-week group, which became more prominent in the recovery phase (Figure 6K). These findings confirm that the persistent fibrosis during the recovery period may not be well reflected by kidney functional parameters.

## 3. Discussion

In the current study, we used a reversible adenine-induced CKD model, adapting it to investigate the dynamic effects on kidney function and injury as well as water and salt homeostasis throughout the phases of disease progression and recovery. The primary findings were that, during the recovery phase from adenine-induced CKD, there was a rapid increase in kidney function, coupled with significant improvement in key water transporter levels and urine osmolality. Moreover, we found a partial reversal in the gene expression of inflammatory markers and the tubular injury marker KIM-1. However, despite these improvements, structural kidney damage and fibrosis persisted even after the recovery period, indicating that the progression of kidney fibrosis might not be reflected by kidney function parameters.

In our study, kidney function, as measured by plasma creatinine, fully recovered during the recovery period. The recovery of kidney function observed in this model is likely due to the clearance of crystals from the tubular lumen, leading to reduced urinary excretion of 2,8-DHA and adenine as well as the compensatory mechanism of the remaining functional nephrons. 

Debate is ongoing about the possibility of reversing severe CKD. Investigating the reversibility of kidney injury is complicated by the fact that most CKD animal models are induced by irreversible injury, such as genetic modification, 5/6 nephrectomy, ischemia–reperfusion injury, or non-reversible ureteral obstruction [15,16]. In this study, we wanted to induce a disease state representative of a severe CKD stage, characterized by a significant decline in kidney function and marked inflammation and fibrosis consistent with previous studies [5,6,8], allowing analyses of the reversibility of renal function, water balance, and renal injury in severe CKD. Our data showed a significant decrease in KIM-1 and several inflammation markers during recovery, suggesting that tubular injury and inflammation are reversible during severe CKD. Nevertheless, the reversibility was only partial as the level of inflammation persisted at a higher magnitude compared to that observed in the healthy control mice. These improvements were less pronounced related to structural kidney damage and fibrosis, demonstrating no significant recovery of collagen deposition, as shown by Sirius Red staining as well as mRNA expression of fibrosis markers. In accordance with these findings, Klinkhammer et al. [17] investigated the potential for recovery in an adenine model. Consistent with our data, they found that the recovery period was characterized by improved kidney function, but histological observations revealed persistent tissue damage, fibrosis, inflammation, and nephron loss. An open question remains whether the duration of the recovery phase in our study was sufficient to detect evidence of the reversal of kidney fibrosis. However, a previous study using the reversible unilateral urethral obstruction (rUUO) model has demonstrated that interstitial fibrosis was present up to 18 months post-UUO reversal [18], emphasizing the persistent nature of fibrosis. Moreover, it is essential to consider the rodent strain when studying rUUO [19]. Research has shown that inbred mouse strains can exhibit varying susceptibility to the development of CKD, with some strains, such as C57BL/6 mice, being susceptible, while others, such as BALB/c mice, are resistant. These strain-dependent differences may also influence the recovery and repair phases of CKD. Interestingly, we found a decrease in the protein levels of FN (as shown by Western blot analysis and immunohistochemistry) during the two-week recovery period, indicating that FN can be subjected to degradation after reversibility of adenine-induced CKD. Matrix metalloproteinases (MMPs) play an important role in the degradation of various ECM components, including FN [20], and we cannot rule out the possibility that MMPs play a role in decreased protein levels of FN during recovery. 

To the best of our knowledge, this is the first study to demonstrate that an adenine-enriched diet for 4 weeks results in an impaired water balance and urine osmolality which significantly improves during the recovery phase. Concurrently, there is a concomitant reduction in AQP2 expression and apical membrane targeting in the collecting ducts, with notably increased membrane targeting returning to control levels after two weeks of recovery. The adenine-induced decrease in AQP2 levels could be due to the signaling properties of adenine through the Ade2R, the expression level of which was increased in the adenine group and started to normalize during the recovery phase. A previous study demonstrated that the Ade2R colocalizes with AQP2 in the collecting duct principal cells [21], indicating that the Ade2R might play an important functional role in the regulation of AQP2. Mechanistically, adenine has been shown to modulate cellular signaling pathways via the Ade2R, such as interfering with the AVP V2 receptor and its downstream cAMP-dependent signaling pathways. Using isolated inner medullary collecting duct cells, it was demonstrated that applying a selective Ade2R antagonist to block Ade2R function could restore the dDAVP-mediated increase in cAMP levels [21], strongly implicating the Ade2R as a mediator of the effects of adenine on AQP2 regulation in collecting duct cells. Thus, modulating the AVP V2 receptor and its downstream cAMP-dependent signaling pathways is crucial for the regulation of AQP2 expression and targeting the plasma membrane leading to water reabsorption [22]. Together, this suggests inhibition of adenine signaling as a potential therapeutic approach. Whether the effect of adenine is direct or indirect remains open. However, one could speculate that adenine may exert direct effects via the Ade2R as well as indirect effects via the adenine metabolite 2,8-DHA, leading to 2,8-DHA nephropathy. In addition, downregulation of AQP1-4 has also been shown in other CKD models [23,24]. This may suggest that downregulation of AQPs can be a recurring feature in experimental kidney diseases driven by inflammatory and fibrotic processes and not exclusively by the direct signaling effect of adenine. In this model, the downregulation of AQPs is therefore most likely the result of a combined effect of the signaling properties of adenine and tubulointerstitial injuries.

Moreover, our data showed that the expression of AQP3 and AQP4 was reduced in response to adenine treatment, contributing to the disturbed water balance. Notably, while AQP3 expression exhibited substantial recovery, AQP4 expression remained diminished even after the 2-week recovery period, indicating that AQP3 may be more susceptible to regulatory mechanisms throughout the recovery phase. This could also reflect that AQP3 might be more important than AQP4 for regulating urine concentration [25]. Our data showed that increased water intake in the 4+2-week group was likely induced by the elevated plasma osmolality, which typically triggers thirst sensation. Simultaneously, improved water reabsorption, due to the increased expression and targeting of AQP2 and AQP3 in this group, could account for the stable urine production despite the higher water intake. However, one would expect that this marked increase in water intake would lead to a more pronounced urine output than we observed. Yet, studies using metabolic cages for only 24 h induce stress in the mice, which leads to variation in the collected data, possibly explaining this discrepancy. 

The downregulation of AQPs in the collecting duct induced by adenine seems to coincide with a simultaneous decrease in AQP1 expression within the proximal tubules. AQP1 has previously been reported to be downregulated after adenine treatment [26], and low levels of AQP1 were shown to exacerbate renal inflammation and fibrosis in adenine-treated mice [12]. These studies revealed that the silencing of AQP1 in cultured proximal tubule cells induced a mesenchymal phenotypic switch [26], while AQP1-KO in mice resulted in reduced kidney function and increased expression of proinflammatory cytokines [12]. Additionally, previous studies have shown that AQP1-null mice display increased tubular injury compared to wild-type mice following ischemia–reperfusion [27] and endotoxin-related acute kidney injury [28], indicating a role of AQP1 in tubular injury and healing. Collectively, these findings suggest that the loss of AQP1 may trigger proximal tubule injury, thus accelerating the progression of kidney disease as we observed in the adenine-induced CKD model. Nevertheless, a Chinese compound (Shen Qi Wan (SQW)) consisting of eight Chinese herbs can activate AQP1 and β-defensin 1 (DEFB1), leading to a reduction in the inflammatory response after adenine treatment [12], indicating a potential therapeutic option of restoring AQP1 expression in CKD. 

Consistent with previous studies, we found that the adenine-treated mice exhibited an increased urinary excretion of sodium related to control mice despite a reduction in food/salt intake [9,29]. This effect aligns with a diminished expression of the sodium transporters NKCC2 and NHE3; however, these transporters remained unaffected during the recovery phase. Altogether, the downregulation of both the water channels (AQP2 and AQP1) as well as sodium transporters (NKCC2 and NHE3) plays an important role in the urinary concentration defect in mice subjected to a 4-week adenine diet. This impaired water and salt homeostasis results in significant volume depletion, as indicated by a sharp reduction in body weight as well as increased BUN and creatinine in the adenine-treated mice. Hence, one may speculate that volume depletion is likely the initial mechanism, which could contribute to later development of renal injury in mice subjected to an adenine-enriched diet. 

We cannot rule out the possibility that dysregulation of the sodium transporters (NHE3 and NKCC2) and AQPs (AQP1-4) may contribute to the overall renal pathology. Previous studies have shown that dysregulation of water and electrolyte transport in the kidney contributes to the progression of fibrosis and inflammation. For example, disrupted sodium transport can lead to tubular injury and activation of pro-inflammatory pathways as well as renal fibrosis [30]. As previously discussed, AQP1 plays a critical role in both tubular injury and the subsequent healing process [28]. Moreover, in the UUO model, RAS blockade with a direct renin inhibitor, aliskiren, restored the protein level of AQP2 by inhibiting the inflammasome, thereby exerting a protective effect [31]. This suggests a connection between AQP2 and the inflammasome, both of which play pivotal roles in renal inflammation. These findings collectively suggest that both sodium transporters and AQPs may play significant roles in the development of renal injury observed in the adenine-induced CKD model. 

The study has several limitations that should be acknowledged: First, the specific molecular mechanisms underlying the responses to adenine-induced CKD and the subsequent recovery, particularly regarding the regulation of the sodium transporters and AQPs, require further investigation. Specifically, it remains unclear whether Ade2R-mediated inhibition of adenylate cyclase directly impacts the expression of all AQP1-4 isoforms, an aspect that requires further investigation to understand its implications on water transport and kidney function during CKD and recovery. Second, our study did not specifically address the changes in Ade2R-mediated inhibition of AVP V2 signaling, including changes in cyclic AMP, activation of protein kinase A, and phosphorylation of AQP2. Exploring these aspects could provide valuable insights into the mechanistic interactions at the cellular level that influence kidney function in this model. Furthermore, the 2-week recovery period evaluated in this study may not provide a comprehensive understanding of the long-term recovery processes of the kidney. Therefore, future research should aim to elucidate the molecular mechanisms in greater detail. Additionally, further studies are warranted to determine whether extending the recovery period could offer a more comprehensive understanding of CKD reversibility. 

In conclusion, we demonstrated that recovery of adenine-induced CKD in mice leads to a rapid increase in kidney function, coupled with significant improvement of key water transporter levels and urine osmolality. Despite these functional improvements, only partial reversal of inflammation, tubular damage, and fibrosis were observed after the recovery period, indicating that the progression of kidney injury might not be reflected by parameters related to kidney function. Further studies are needed to provide a more comprehensive understanding of the underlying mechanism governing the reversibility observed in this model. These future studies should aim to elucidate the processes and molecular pathways that contribute to the recovery of kidney function. By elucidating these mechanisms, we can gain insights into the factors that enable reversibility in the context of adenine-induced CKD. 

## 4. Materials and Methods

### 4.1. Animal Experiments

All procedures were conducted in accordance with the Danish Animal Welfare Act. Experimental protocols were approved by both the local and national ethical review committees (Danish Animal Experiments Inspectorate) and carried out in compliance with the Project License (No. 2020-15-0201-00617).

Eight-week-old C57BL/6J male mice were used in the experiment. The mice were kept in cages with a 12:12 h artificial light cycle and controlled temperature and humidity. All mice had free access to water and the assigned diet. Prior to the study’s start, all mice were allowed acclimatization to the animal facility conditions during a 7-day period. 

The adenine diet used in this experiment consisted of 0.2% adenine and 6% casein (Altromin, Lage, Germany), which has been reported to blunt the taste and smell of adenine [6]. However, the casein did not adequately disguise the adenine, and we therefore covered the pellets in a hazelnut spread to further enhance the concealment of adenine. The amount of hazelnut spread was matched between the adenine groups and the control group.

The mice were randomly divided into four groups: The first group (control) was fed the same casein diet for up to 6 weeks but without the addition of adenine to exclude the possibility of an impact of casein on kidney function (*n* = 11 mice). Control mice were sacrificed at 4, 5, and 6 weeks (*n* = 2–4 per time point). Since there were no significant differences in plasma creatinine levels and BUN across these time points, we decided to pool the data from all control mice for further analysis. The second group was fed the adenine-containing diet for 4 weeks (*n* = 8 mice). The third (*n* = 8 mice) and fourth (*n* = 7 mice) groups were fed the adenine-containing diet for 4 weeks and then switched to the casein diet for 1 and 2 weeks, respectively. Body mass and food and water intake were monitored daily from day 0 to 16 and every other day for the remainder of the experiment. Mice were placed individually in metabolic cages for 24 h before starting the respective diets and again for 24 h one day before sacrifice (*n* = 5–7 mice). One mouse was euthanized after four weeks of adenine feeding due to severe weight loss.

At sacrifice, anesthesia was induced with 5% sevoflurane (maintained at 3–4%), and the mouse was placed on a heating pad. Blood was collected through cardiac puncture, and immediately after, both kidneys were excised and weighed. The left kidney was dissected into inner medulla and cortex for subsequent RNA and protein assessment, while the right kidney was fixed in 4% formaldehyde for histological examination. 

### 4.2. Plasma and Urine Biochemistry

Plasma creatinine levels were determined using a Creatinine Assay Kit (Sigma Aldrich, St. Louis, MI, USA), according to the manufacturer’s instructions (*n* = 7–11 mice). Blood urea nitrogen (BUN) levels were determined using a Urea Nitrogen Colorimetric Detection Kit (Thermo Fisher Scientific, Waltham, MA, USA), according to the manufacturer’s instructions (*n* = 7–11 mice). Plasma and urine osmolality were determined by freeze-point depression (Osmomat 030-D, Gonotec, Bie and Berntsen, Herlev, Denmark) (*n* = 5–11 mice). 

Sodium and potassium concentrations in plasma and urine were measured using a model 420 flame photometer (Sherwood Scientific, Cambridge, UK) (*n* = 5–11 mice). Proteinuria was measured using the Pierce BCA Protein Assay Kit (Roche, Basel, Switzerland).

### 4.3. Quantitative PCR

Cortex and inner medullary tissue were homogenized using a TissueLyzer LT (Qiagen, Hilden, Germany). RNA was purified using a NucleoSpin^®^ RNA kit (Macherey Nagel, Düren, Germany) according to the manufacturer’s instructions. RNA concentration was quantified by measuring the optical density at 260 nm using a BioPhotometer 6131 (Eppendorf, Hamburg, Germany). cDNA was synthesized using a RevertAid First Strand synthesis kit #K1622 (Thermo Fisher Scientific, Waltham, MA, USA). Preparations of qPCR samples were performed with a Maxima SYBR Green qPCR Master Mix (Thermo Fisher Scientific, Waltham, MA, USA), following the manufacturer’s instructions. The reaction was run on an Aria Mx3000P qPCR System (Agilent Technologies, Santa Clara, CA, USA). The mRNA expression for each signal was calculated by using the ΔCt procedure, with GAPDH as the reference gene (*n* = 7–11). The primer sets are summarized in Appendix A. 

### 4.4. Immunoblotting

The procedure was similar to what has been described in detail previously. Briefly, cortex tissue was homogenized in RIPA buffer containing the protease inhibitors phosphatase inhibitor cocktails 2 and 3 (Sigma Aldrich, St. Louis, MO, USA) and complete mini protease inhibitor cocktail tablets (serine, cysteine, and metalloprotease inhibitor, Roche, Hvidovre, Denmark) using a TissueLyser LT (Qiagen, Hilden, Germany). Afterwards, samples were centrifuged, and the supernatant was assayed for protein concentration (Pierce BCA Protein Assay Kit, Roche). Gel samples were prepared in Laemmli buffer containing 2% SDS. Total protein was separated by SDS/PAGE using 12% Criterion TGF stain-free gels (Bio-Rad Laboratories, Copenhagen, Denmark) and transferred to a nitrocellulose membrane (Bio-Rad Laboratories, Copenhagen, Denmark). Subsequently, blots were blocked with non-fat dry milk in PBS-T (80 mM Na_2_HPO_4_, 20 mM NaH_2_PO_4_, 100 mM NaCl, 0.1 Tween 20, adjusted to pH 7.4). The blots were then incubated with primary antibodies overnight at 4 °C. The primary antibodies were visualized with horseradish-peroxidase-conjugated secondary antibodies and Pierce ECL Western Blotting Substrate. Antibodies are summarized in Appendix A. All Western blots were normalized to total protein, as measured using stain-free technology (*n* = 7–11) [32].

### 4.5. Histology and Immunohistochemistry

Following removal, the right kidney was immersed in 4% PFA for one hour, rinsed with PBS, dehydrated in a series of alcohol, and embedded in paraffin. Tissue sections (2 µm) were deparaffinized, rehydrated, and rinsed (*n* = 5). Then, they were blocked with 35% hydrogen peroxide (H_2_O_2_) in methanol for 30 min. For epitope retrieval, sections were boiled in a target retrieval solution (1 mM Tris, 0.5 mM EGTA, pH of 9.0) for 10 min then left to cool and incubated in a shielding buffer (50 mM NH_4_Cl in PBS) for 30 min. Sections were then blocked in a blocking buffer (PBS containing 1% BSA, 0.2% gelatin, and 0.05% saponin) and incubated with primary antibodies diluted in antibody staining solution (PBS containing 0.1% BSA and 0.3% Triton X100) for 1 h at room temperature in a humidity chamber, followed by overnight incubation at 4 °C (see Appendix A for specificity and dilution). The following day, the sections were rinsed three times, followed by incubation with secondary antibodies diluted in antibody staining solution (Appendix A) for 1 h at room temperature. Afterward, sections were rinsed and incubated with 3,3′-diaminobenzidine tetrachloride (DAB) dissolved in water containing 0.1% H_2_O_2_ to visualize the sites of antibody–antigen reactions. Additionally, 3 µm sections were stained with hematoxylin-eosin (H&E) (*n* = 5) and Sirius Red (*n* = 5). For H&E staining, the rehydrated sections were stained with Mayer’s hematoxylin solution for 5 min, rinsed, and stained with Eosin Yellow solution for 30 s before being dehydrated and mounted. For Sirius Red staining, the rehydrated sections were left in tap water for 10 min and stained for 30 min before being dehydrated and mounted. The sections were imaged using an Olympus VS120 Virtual Slide Scanner with a CCD color camera (AVT Pike F-505C VC50 progressive scan CCD color camera) with a 24-bit setting. A 40× air objective lens (NA 0.95) was used. The images were stitched automatically.

### 4.6. Immunofluorescence Staining and Imaging

Immunofluorescence (IF) staining of α-SMA and PDGFRβ was performed using tissue sections mounted on microscope slides, while IF staining of AQP2, KIM-1, NKCC2, and AQP1 was performed in a multiplex fashion using coverslip-mounted paraffin-embedded tissue sections, a method described by Elsborg et al. [33]. Briefly, 25 mm round coverslips were cleaned and coated in a gelatin coating solution and dried for 48 h at room temperature. Kidney sections (2 µm) were placed onto the coated coverslips or SuperFrost+ microscope slides. The sections were deparaffinized in xylene and rehydrated in a graded series of ethanol. Afterwards, the sections were boiled in a target retrieval solution for 10 min and incubated in the shielding buffer for 30 min. The sections were blocked in blocking buffer and then incubated with primary antibodies overnight at 4 °C. The day after, the sections were incubated with secondary antibodies and Hoechst (33 342; Thermo Fisher Scientific, Waltham, MA, USA) for 50 min at room temperature and kept dark. Images of the coverslip-mounted kidney sections were acquired on a Nikon Ti Eclipse inverted fluorescence microscope equipped with a CoolLED pE-300 unit. The acquisition was performed with an Andor Zyla 5.5 Mpixel camera (Andor Technology Ltd., Belfast, UK) with a 12-bit setting. The image visualization utilized excitation/emission filter cubes with wavelengths of 370/460 nm, 460/510 nm, and 550/670 nm. The objective lens used was a 60× oil objective (NA 1.40). The images were stitched automatically. For imaging of α-SMA and PDGFRβ, the microscope slides were mounted on coverslips and acquired using an Olympus VS120 Virtual Slide Scanner with a Hamamatsu Orca-Flash 4.0 CMOS 4 Mpixel camera with a 16-bit setting. The excitation/emission filter cubes for DAPI, FITC, and Cy5 and the objective lens used was 40× air objective (NA 0.95). The images were stitched automatically. Antibodies are summarized in Appendix A. For multiplexing, antibodies and Hoechst were removed from the tissues between each staining round using a mild stripping buffer (0.5 M Tris base, 10% SDS and 1 M Dithiothreitol (DTT) in deionized water). After stripping, if no remaining staining was detectable in the microscope, the coverslips were rinsed, and the sections entered another staining cycle starting from the blocking step. The staining sequence was as follows: First, AQP2 and KIM-1 were imaged using the FITC filter and Cy5 filter, respectively. Second, NKCC2 was imaged using the Cy5 filter, and third, AQP1 was imaged using the FITC and Cy5 filter (*n* = 5).

### 4.7. Quantitative Image Analysis

Image analysis of IF stainings was performed using Fiji [34], while immunohistochemistry and Sirius Red staining images were analyzed using QuPath (V.0.4.3) image analysis software [35].

To quantify the area of FN and Sirius Red staining, first representative training images were loaded into QuPath for training of pixel classifiers. One pixel classifier was generated recognizing the tissue and a second was made to recognize the FN/Sirius Red staining. Afterwards, the inner medulla and the cortex were annotated manually, and the area covered by the staining was calculated in both the cortex and medulla and expressed relative to the total area of tissue.

AQP1- and KIM-1-positive areas were determined similarly. First, the cortical area was manually selected. Hereafter, a threshold-dependent mask was created to segment positive tubules from negative kidney tubules. This threshold was based on the average fluorescence intensity within positive tubules. The coverage of the mask within the annotated cortex was calculated and expressed as a percentage of the positive tubular area within the cortical areas of the kidneys.

To quantify the subcellular localization of AQP2, we measured the AQP2 protein expression levels from the basal to the luminal side of the principal cells in the inner medulla, as described previously [36]. Briefly, a line (10 pixels wide and 38 pixels long) was drawn across 15 cells from 5 mice per group, resulting in 75 lines per group. A line scanning analysis was made, where the fluorescence intensity was measured across 41 points along each line, creating a gradient of fluorescence intensity of the AQP2 staining across AQP2-positive cells. 

To measure NKCC2 levels, the outer stripe of outer medulla was manually selected using the free hand selection tool. Afterwards, the background was subtracted, and the mean fluorescence intensity was measured within the annotated area.

### 4.8. Adenine and 2,8-DHA Measurements

Adenine and 2,8-DHA concentrations were measured in plasma and urine samples using liquid chromatography tandem mass spectrometry (LC-MS/MS) (*n* = 5–11 mice). First, 50 µL of plasma and urine were mixed with 200 µL ice-cold methanol. The samples were vortexed at 1000 rpm for 5 min with 10 s cooling on ice every minute and incubated at least overnight at −20 °C. The samples were then centrifuged at 15,000× *g* for 20 min at 5 degrees, and 200 µL of the supernatant (80% of the total volume) was transferred and vacuum dried. The dried samples were reconstituted in 40 µL (80% of the starting material) of LC buffer containing LC-MS-grade water and 0.2% formic acid. The samples were then mixed for 5 s using ultrasonication and stored at −20 °C until use. The samples were analyzed by LC-MS/MS on a Vanquish Horizon LC coupled with a Q Exactive Plus Orbitrap MS (both from Thermo Fisher Scientific, Waltham, MA, USA) in positive mode with settings as previously described [37]. Peak integration of the parent masses ([M+H]^+^) was performed in Skyline version 21.2 (MacCoss Lab, University of Washington, Seattle, WA, USA) [38]. Standard ranges were generated by serial dilutions to obtain eight concentration levels. The standard range for adenine (Sigma Aldrich, St. Louis, MO, USA) ranged from 1 nM to 100 µm and for 2,8-DHA (Cayman Chemicals, Ann Arbor, MI, USA) from 3.4 nM to 3.4 mM. Both standard ranges were prepared and analyzed using identical conditions for the urine and plasma samples. The slopes of the calibration curves were generated on log10-transformed peak integration data and the concentration of adenine and 2,8-DHA were determined by linear regression analysis.

### 4.9. Statistical Analysis

Data are presented as means ± SEM unless otherwise stated. Multiple comparisons between experimental groups were performed using a one-way ANOVA, followed by Tukey’s multiple comparisons test. Baseline and terminal urine output and osmolality were compared using a paired *t*-test. GraphPad Prism software V.10.2.1 (GraphPad Software, La Jolla, CA, USA) was used for all statistical analyses. *p* < 0.05 was considered significant.

## Figures and Tables

**Figure 1 ijms-25-03447-f001:**
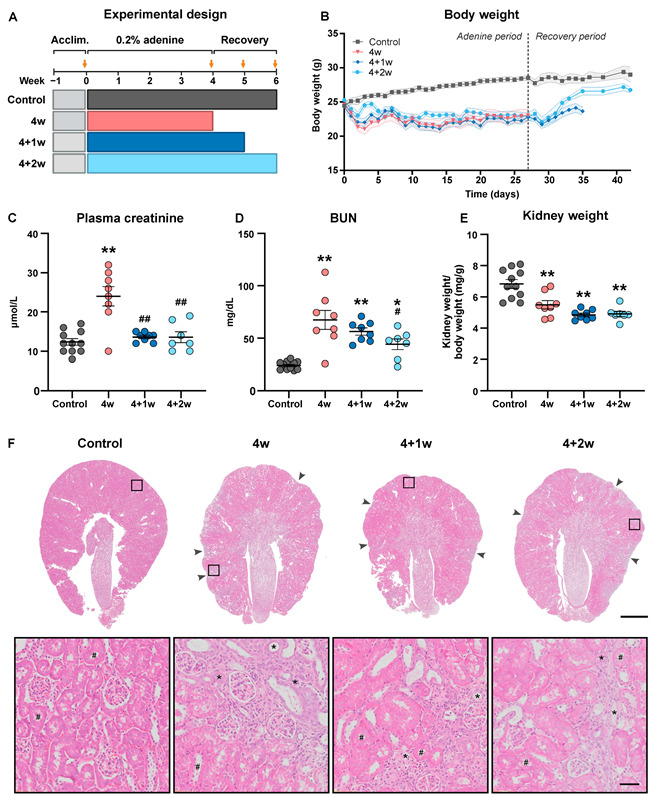
Kidney function was recovered despite persistent kidney weight loss and scarring in adenine-fed mice. (**A**) Experimental design showing the timeline and groups of mice receiving 0.2% adenine for 4 weeks (4w) followed by a recovery period of either 1 or 2 weeks (4+1w and 4+2w). Yellow arrows indicate time points where mice were kept in metabolic cages. (**B**) Body weight over the study period. The dashed vertical line indicates the transition from adenine diet to control diet. *p* < 0.05 for days 1–28 (4w), days 1–35 (4+1w), and day 2 as well as days 5–33 (4+2w) compared to controls (*n* = 7–11 mice). (**C**) Plasma creatinine (*n* = 7–11 mice). (**D**) Blood urea nitrogen (BUN) (*n* = 7–11). (**E**) Mean kidney weight relative to body weight (*n* = 7–11 mice). (**F**) Representative microscopy images of kidney tissue following H&E staining. Arrowheads indicate loss of functional tubule mass. Scale bar: 1000 µm. Representative zoomed-in views are inserted in the bottom panel. Atrophic tubules are indicated by asterisks (*) and healthy tubules by hash marks (#). Scale bar: 50 µm. Values are presented as mean ± SEM. * *p* < 0.05, ** *p* < 0.01 compared to controls and ^#^
*p* < 0.05, ^##^
*p* < 0.01 compared to the 4w group.

**Figure 2 ijms-25-03447-f002:**
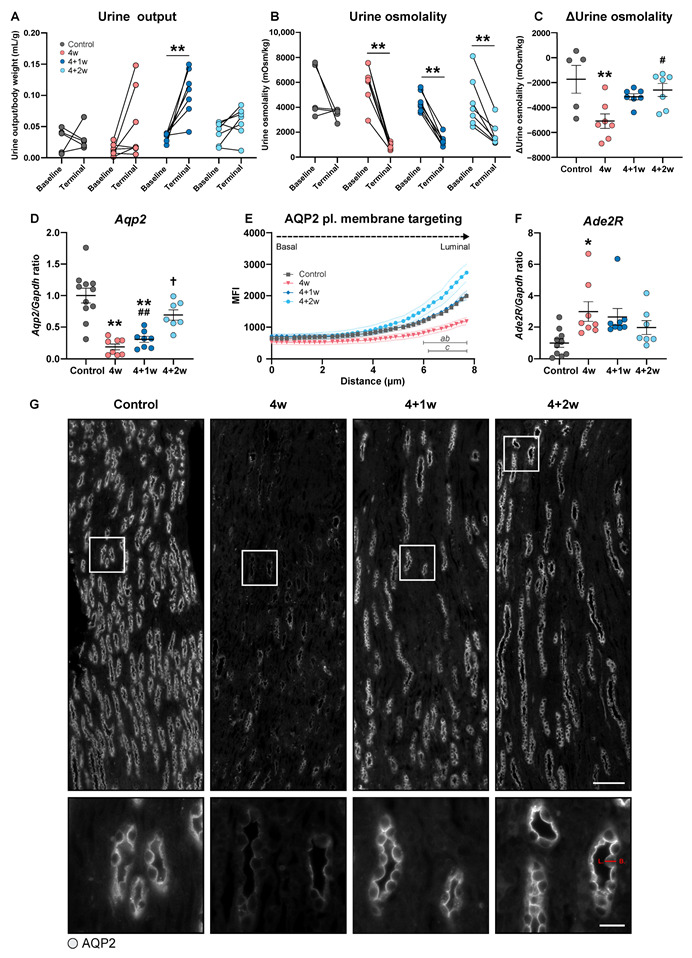
Recovery of water balance and urine osmolality are closely associated with increased AQP2 expression and plasma membrane targeting. (**A**) Urine output measured 24 h before experimental start (baseline) and 24 h before sacrifice (terminal) (*n* = 5–7 mice). (**B**) Urine osmolality measured 24 h before experimental start (baseline) and 24 h before sacrifice (terminal) (*n* = 5–7 mice). (**C**) Difference between baseline and terminal urine osmolality (*n* = 5–7 mice). (**D**) Gene expression of *Aqp2* in inner medulla (*n* = 7–11 mice). (**E**) AQP2 plasma (pl.) membrane targeting in inner medulla. Mean fluorescence intensity (MFI) of the AQP2 localization along the longitudinal axis of the principal cells in inner medulla. (**F**) Gene expression of Ade2R in inner medulla (*n* = 7–11 mice). (**G**) Representative immunofluorescence images from renal inner medulla of AQP2. Scale bar: 100 µm. White squares are inserted in the bottom panel. Scale bar: 10 µm. The red line represents the full length of the principal cell from the basal (B) side to the luminal (L) side along which the MFI was measured. Values are presented as mean ± SEM. * *p* < 0.05, ** *p* < 0.01 compared to controls and ^#^
*p* < 0.05, ^##^
*p* < 0.01 compared to 4w, ^†^
*p* < 0.05 compared to 4+1w. a: *p* < 0.05 4+1w vs. 4w; b: *p* < 0.05 4+2w vs. 4w; c: *p* < 0.05 4w vs. control.

**Figure 3 ijms-25-03447-f003:**
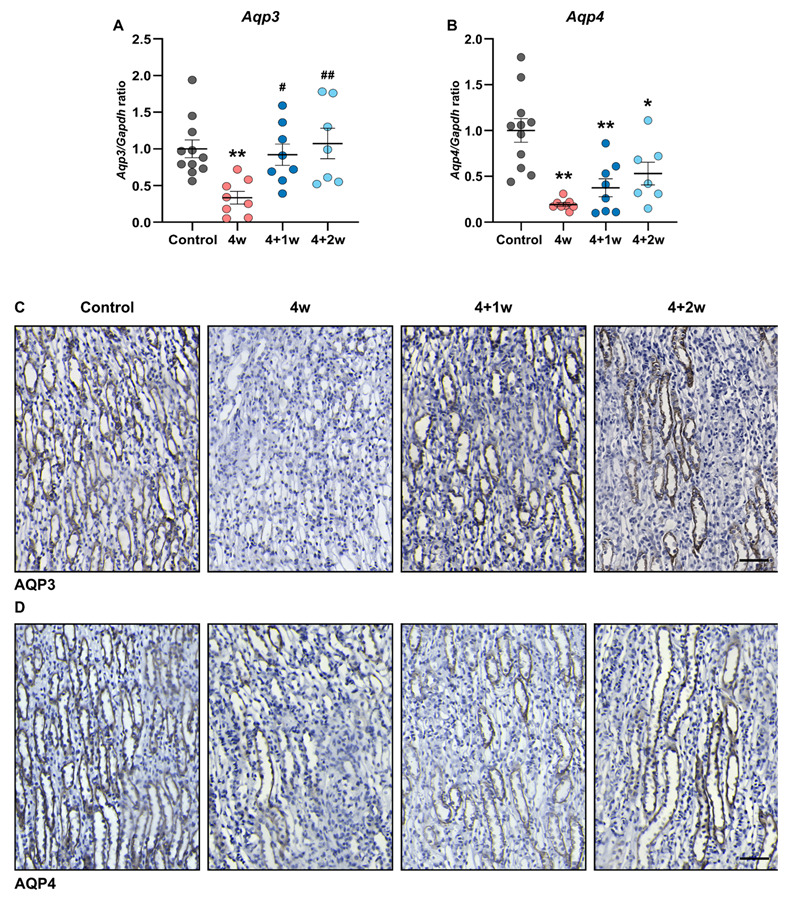
Adenine feeding reduces AQP3 and AQP4 expression in the inner medulla. Gene expression of (**A**) *Aqp3* and (**B**) *Aqp4* in inner medulla (*n* = 7–11 mice). Representative immunohistochemistry images of (**C**) AQP3 and (**D**) AQP4 in the basolateral membrane of principal cells in the inner medulla. Scale bars: 50 µm. Values are presented as mean ± SEM. * *p* < 0.05, ** *p* < 0.01 compared to controls and ^#^
*p* <0.05, ^##^
*p* < 0.01 compared to 4w.

**Figure 4 ijms-25-03447-f004:**
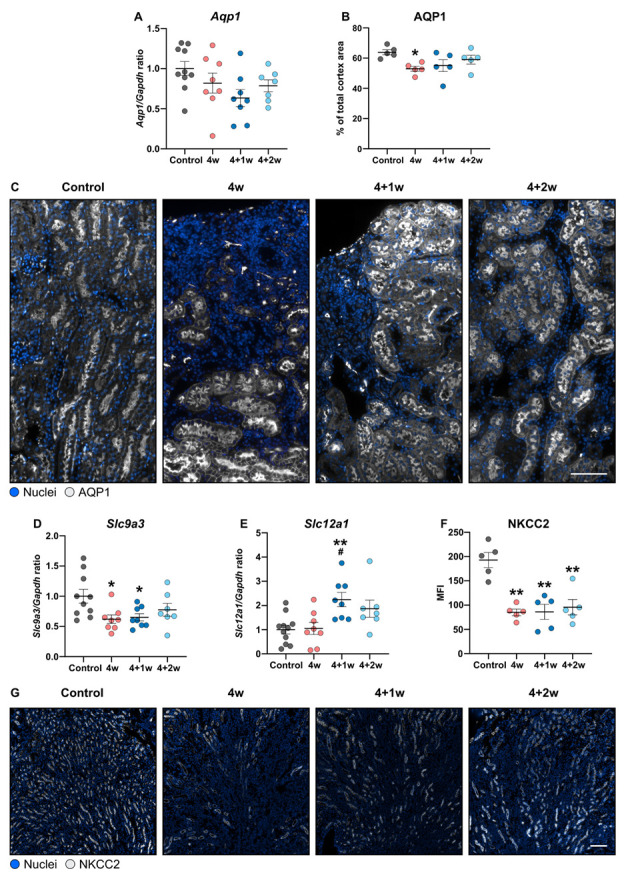
Recovery of adenine-induced CKD does not affect the expression of AQP1, NHE3, and NKCC2. (**A**) Gene expression of *Aqp1* in cortex (*n* = 7–11 mice). (**B**) Percentage of positive area of AQP1 in the cortical region (*n* = 5 mice) and (**C**) representative immunofluorescence images from renal cortex of AQP1. Scale bar: 1000 µm. Gene expression of (**D**) *Slc9a3* in cortex and (**E**) *Slc12a* in medulla (*n* = 7–11 mice). (**F**) Mean fluorescence intensity (MFI) of NKCC2 within the mTAL (*n* = 5 mice) and (**G**) representative immunofluorescence images from renal outer medulla of NKCC2. Scale bar: 100 µm. Values are presented as mean ± SEM. * *p* < 0.05, ** *p* < 0.01 compared to controls and ^#^
*p* < 0.05 compared to the 4w group.

**Figure 5 ijms-25-03447-f005:**
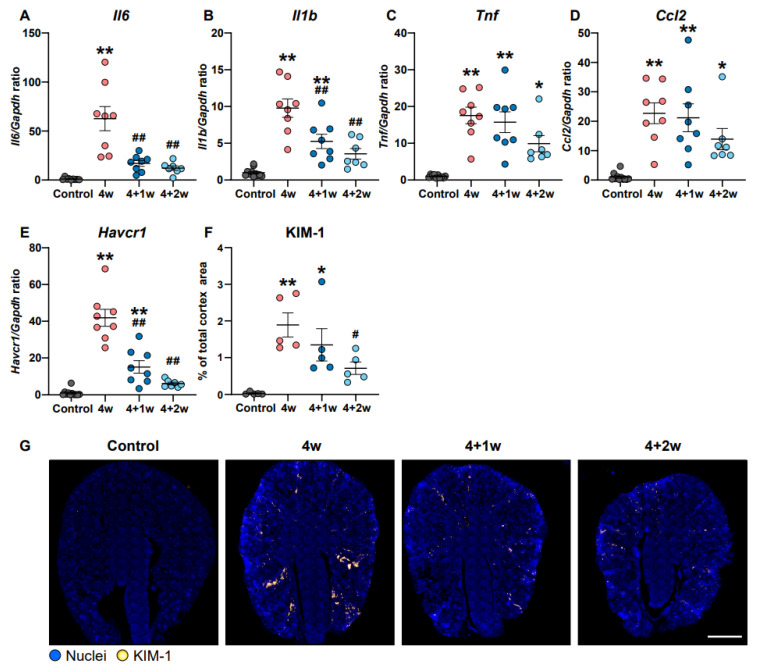
Kidney inflammation and injury in the cortex is partially restored following recovery of adenine-induced CKD. Gene expression of (**A**) interleukin 6 (*Il6*), (**B**) interleukin 1 beta (*Il1b*), (**C**) tumor necrosis factor (*Tnf*), (**D**) C-C motif chemokine ligand 2 (*Ccl2*), and (**E**) hepatitis A virus cellular receptor 1 (*Havcr1*) in cortex (*n* = 7–11 mice). (**F**) Percentage of positive area of kidney injury molecule-1 (KIM-1) in cortex (*n* = 5 mice). (**G**) Representative immunofluorescence images of KIM-1. Scale bar: 1000 µm. Values are presented as mean ± SEM. * *p* < 0.05, ** *p* < 0.01 compared to controls and ^#^
*p* < 0.05, ^##^
*p* < 0.01 compared to the 4w group.

**Figure 6 ijms-25-03447-f006:**
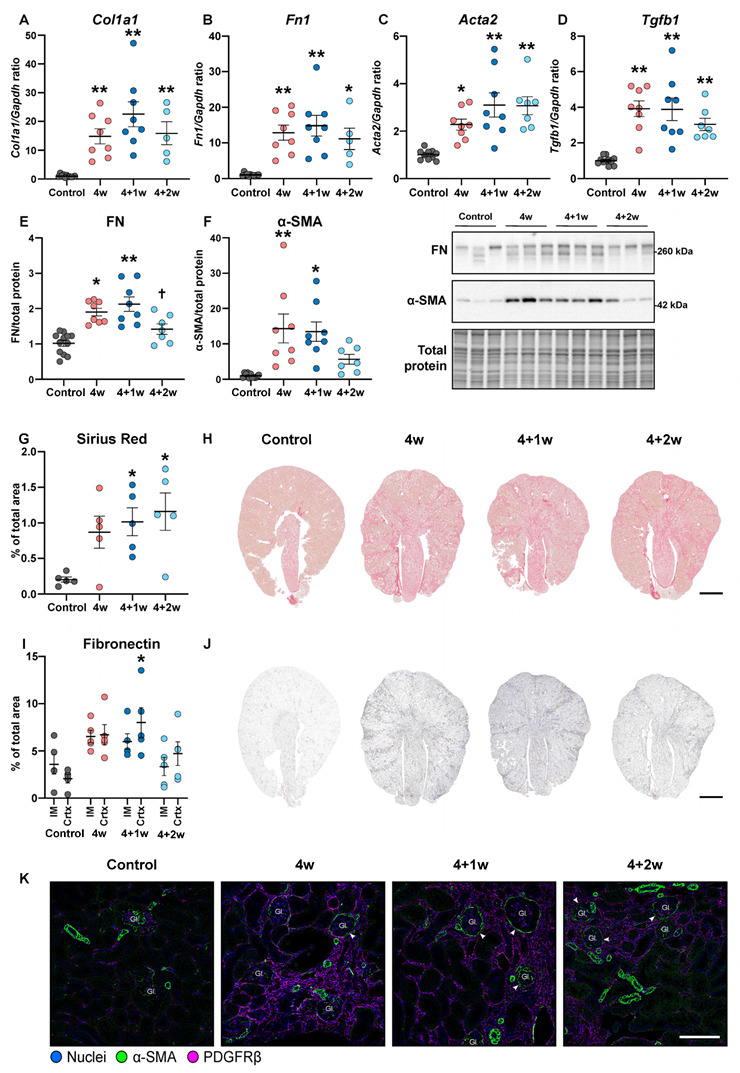
Kidney fibrosis is persistent following recovery. Gene expression of (**A**) collagen 1a1 (*Col1a1*), (**B**) fibronectin (*Fn1*), (**C**) α-smooth muscle actin (*Acta2*), and (**D**) transforming growth factor-β (*Tgfb1*) in cortex (*n* = 5–11 mice). Western blotting analyses of (**E**) fibronectin (FN) and (**F**) α-smooth muscle actin (α-SMA) from cortex (*n* = 7–11 mice). (**G**) Quantification of Sirius Red staining from the whole kidney (*n* = 5 mice) and (**H**) representative microscopy images. Scalebar 1000 µm. (**I**) Quantification of fibronectin (FN) immunohistochemistry from cortex and inner medulla (*n* = 5 mice) and (**J**) representative microscopy images. Scale bar 1000 µm. (**K**) Representative immunofluorescence images from renal cortex of α-SMA and platelet-derived growth factor receptor-β (PDGFRβ). Gl. = Glomerulus. Arrowheads show accumulation of α-SMA in periglomerular cells. Scale bar: 100 µm. Values are presented as mean ± SEM. * *p* < 0.05, ** *p* < 0.01 compared to controls and ^†^
*p* < 0.05 compared to 4+1w.

**Table 1 ijms-25-03447-t001:** Food and water intake before and after adenine treatment.

	Control	4w	4+1w	4+2w
**Food intake (g)**	4.35 ± 0.41	1.38 ± 0.24 **	5.06 ± 0.22 ^##^	4.91 ± 0.29 ^##^
**Water intake (mL)**	4.96 ± 0.33	4.29 ± 0.87	5.86 ± 0.67	8.11 ± 0.72 *^##^
**Urine**				
Adenine (µM)	1.14 ± 0.014	99.40 ± 19.87 **	0.15 ± 0.070 ^##^	0.58 ± 0.49 ^##^
Adenine (µg/24 h)	0.11 ± 0.008	18.29 ± 6.55 *	0.050 ± 0.021 ^##^	0.037 ± 0.017 ^##^
2,8-DHA (µM)	0.070 ± 0.026	19.84 ± 3.87 **	0.19 ± 0.045 ^##^	0.31 ± 0.13 ^##^
2,8-DHA (µg/24 h)	0.022 ± 0.008	6.20 ± 1.21 **	0.059 ± 0.014 ^##^	0.096 ± 0.040 ^##^

Values are presented as mean ± SEM (*n* = 5–7 mice). * *p* < 0.05, ** *p* < 0.01 compared to controls. ^##^ *p* < 0.01 compared with 4w group.

**Table 2 ijms-25-03447-t002:** Blood and urinary parameters.

	Control	4w	4+1w	4+2w
**Plasma**				
Osmolality (mOsm/kg)	315.5 ± 1.6	320.6 ± 1.4	320.5 ± 1.9	324.9 ± 2.7 **^#^
Na^+^ (mM)	151.8 ± 1.6	150.0 ± 1.7	157.6 ± 2.3 ^#^	158.9 ± 2.2 ^#^
K^+^ (mM)	7.7 ± 0.56	6.8 ± 0.50	8.3 ± 0.91	7.2 ± 0.75
**Urine**				
Osmolality (mOsm/kg)	3482 ± 210	856.6 ± 101 **	1284 ± 1778 **	1810 ± 366 **^#^
Na^+^ (mM)	198.2 ± 13.8	71.0 ± 15.0 **	68.5 ± 13.4 **	117.0 ± 31.3
Na^+^ excretion (Na^+^ excretion/intake)	0.44 ± 0.07	0.52 ± 0.08	0.34 ± 0.02	0.33 ± 0.03
K^+^ (mM)	208.8 ± 17.3	77.7 ± 23.8 **	61.9 ± 39.0 **	108.8 ± 27.4 *
K^+^ excretion (K^+^ excretion/intake)	0.17 ± 0.02	0.20 ± 0.06	0.11 ± 0.01	0.12 ± 0.01
Proteinuria (mg/24 h)	28.3 ± 7.4	17.3 ± 5.7	23.6 ± 2.9	24.3 ± 3.1

Values are presented as mean ± SEM (*n* = 5–11 mice). * *p* < 0.05, ** *p* < 0.01 compared to controls. ^#^
*p* < 0.05 compared to the 4w group.

## Data Availability

The original contributions presented in the study are included in the article; further inquiries can be directed to the corresponding author.

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
