# Peer review of "Recovery of Water Homeostasis in Adenine-Induced Kidney Disease Is Mediated by Increased AQP2 Membrane Targeting"

_ijms, 2024, doi:10.3390/ijms25063447_

Round 1

Reviewer 1 Report

Comments and Suggestions for Authors

In this article, the authors studied a reversible CKD model induced by an adenine diet for 4 weeks, followed by a two-week recovery period. During treatment, BUN increased with a decrease in urine osmolality, and there was a reduction in the expression of AQP1, AQP2, NHE3, and NKCC2. Moreover, kidney inflammation and injury markers in the cortex, as well as fibrosis, were upregulated. Many of these parameters were reversed during the 2-week recovery period. The article presents some novelty, particularly regarding the phenotypes observed during the recovery period. However, the mechanisms underlying this improvement during the recovery period were not elucidated. The following concerns need to be addressed:

  1. No experiments were conducted to demonstrate whether adenine could directly inhibit AQP2 via AdeRs.

  2. What is the expression of other AQPs? Do they (other than AQP2) affect the phenotype of mice on an adenine diet in terms of inflammation and fibrosis?

  3. The authors studied a CKD model that works well in rats. Similar studies have been conducted before in rats (Ingrid F. Dos Santos et al. American Journal of Physiology-Renal Physiology 2019 316:4, F743-F757). Although the 2-week recovery model has some novelty, why the authors chose a 2-week recovery period was not discussed.

  4. The authors claim that kidney function declined after 4 weeks without urinary albumin/creatinine ratio or GFR.

  5. The authors should explain how water intake did not increase during treatment but water output increased many-fold (Table 1, Figure 2A). Moreover, water intake doubled during the recovery period without increasing urinary output.

  6. Statistics for Fig 3B are missing.

  7. For Fig 3C, authors should choose pictures displaying approximately an equal number of nuclei.

  8. Figures 4F and G do not correspond.

  9. In Fig 5 E/F, authors should show a loading control/housekeeping gene for western blots.

  10. The location of the kidney is missing in all histology figures.

  11. Authors should explain the improvement in urinary sodium during recovery.

Reviewer 2 Report

Comments and Suggestions for Authors

I have had the opportunity to thoroughly review the manuscript with the identifier ijms-2872533, which comprehensively contributes to understanding CKD recovery mechanisms, and AQP2's role is recognized, with potential implications for new therapeutic approaches. Here are some comments from the article:

1. Clarification on Fibrosis Reversibility: While the manuscript highlights the persistence of fibrosis despite the functional recovery, a more detailed exploration of why fibrosis remains resistant to reversal and how this aligns or contrasts with other models of CKD would enhance the reader's understanding.

2. Broader Implications: The discussion could benefit from a more explicit connection between the study's findings and their implications for CKD treatment strategies. Specifically, how the insights into AQP2's role and adenine signaling might inform therapeutic approaches.

3. Limitations and Future Directions: While the discussion touches on the partial reversibility of inflammation and tubular damage, explicitly stating the study's limitations and proposing specific future research directions based on the findings would provide clarity and a pathway forward for the field.

4. Role of Other Transporters and Channels: The discussion on the downregulation of AQP1 and sodium transporters (NKCC2 and NHE3) is insightful, but integrating a discussion on how these changes interact with AQP2 alterations and contribute to the overall renal pathology could offer a more comprehensive view.

5. In Figure 1F, a zoomed-in view could be included to observe the loss of functional tubule mass more clearly.

Reviewer 3 Report

Comments and Suggestions for Authors

The aim of this manuscript is to investigate a reversable CKD mouse model and potential mechanism. The author using adenine-enriched diet model along with 2 weeks recovery period and showed that significantly impaired renal function after 4 weeks adenine-enriched diet period. They also found partially improvement of renal function during recovery period.

1. This animal model is not new, and has been reported since 2016. Is there any modification from the previous model?

2. The study groups are little confused. For the controll group, how long was the control group being studied? 4 weeks or 6weeks?

3. The author seems to be replicating the animal model with detailed biochemical markers, such as more fibrosis markers, inflammation markers and so on. The real connection between AQP1 and renal function is not very clear.

4. What is the potential therapeutic option based on this finding? Restore AQP1 expression?

Round 2

Reviewer 1 Report

Comments and Suggestions for Authors

The manuscript has been significantly improved by the authors in the current revision. However, following questions remained unanswered.

1. This article argue for the possibility of reversing severe CKD. Not sufficient evidence of developing CKD or kidney function decline after Adenine diet.
Authors performed protein measurement in urine with a BCA measurement kit which does not address total urinary albumin or ACR ratio.

2. The revised manuscript presents intriguing data regarding the decrease in AQP 1-4 expression. However, further investigation is needed to explore the following questions:

  • Does Ade2R-mediated inhibition of adenylate cyclase directly affect the expression of all AQP1-4 isoforms?
  • Are there changes in cAMP levels observed in kidney tissue during CKD and the recovery period?
  • Is there evidence of co-localization between Ade2R and AQP 1-4 proteins?

3. While the manuscript focus on the recovery of water homeostasis, the observed variability in experimental data (water intake/urine output) raises concerns about the overall strength of the conclusions. Moreover, the title mentions "AQP2 Membrane Targeting," but the study identifies the involvement of other AQPs as well. Further clarification is needed to determine whether these effects are direct or indirect consequences of the treatment.

By addressing these points, the authors can strengthen the manuscript and provide a more comprehensive understanding of their findings.

Author Response

This article argue for the possibility of reversing severe CKD. Not sufficient evidence of developing CKD or kidney function decline after Adenine diet. Authors performed protein measurement in urine with a BCA measurement kit which does not address total urinary albumin or ACR ratio.

We appreciate the emphasis on the importance of specific markers for kidney function and damage, such as urinary albumin or the albumin-to-creatinine ratio (ACR). We agree with the reviewer that a BCA protein kit does not address urinary albumin. However, acquiring an ELISA kit for direct albumin measurements in mice within the timeframe for these revisions was unfortunately not feasible. 

As previously shown, 0.2% adenine feeding for 4 weeks in C57BL/6 mice does not lead to the development of proteinuria, which is consistent with our findings [1, 2]. Although multiple studies have found development of proteinuria and albuminuria in rats fed an adenine-enriched diet [3-5]. It has previously been demonstrated that the C57BL/6 mouse strain is resistant to the development of proteinuria and albuminuria compared to other strains [6-9].

The revised manuscript presents intriguing data regarding the decrease in AQP 1-4 expression. However, further investigation is needed to explore the following questions:

  • Does Ade2R-mediated inhibition of adenylate cyclase directly affect the expression of all AQP1-4 isoforms?
  • Are there changes in cAMP levels observed in kidney tissue during CKD and the recovery period?
  • Is there evidence of co-localization between Ade2R and AQP 1-4 proteins?

We thank the reviewer for this comment. We agree that further investigation is needed for deeper exploration into the impact of Ade2R-mediated signaling on the expression of AQP1-4 isoforms and in our opinion; this is beyond the scope of this work.

Potential analysis of cAMP levels in the tissue would be a global analysis on tissue from a new set of experimental animals. We do not think such a global analysis would be informative regarding cAMP levels in the cells expressing the different AQPs. 

Recognizing the significance of these questions for advancing our understanding of the mechanisms at play, we have incorporated these considerations into the discussion section of our revised manuscript. This inclusion aims to highlight the potential implications of Ade2R-mediated signaling on AQP1-4, strongly motivating directions for future research, and acknowledging the limitations of our current study.

In response to the inquiry about the co-localization of Ade2R with AQP1-4, it is important to note that Ade2R primarily localizes to the apical domain of collecting duct cells so to the same cells as AQP2, AQP3 and AQP4 [10]. Thus, on the subcellular level, Ade2R does not co-localize with AQP3 and AQP4 in collecting duct principal cells nor with AQP1 in proximal tubule cells. AQP2 is in subapical intracellular vesicles as well as the apical plasma membrane and as such in the same subcellular domain as Ade2R. However, since the size of the AQP2 vesicles are below the resolution of light it would require immune-EM to investigate which we believe is beyond the scope of this paper.

While the manuscript focus on the recovery of water homeostasis, the observed variability in experimental data (water intake/urine output) raises concerns about the overall strength of the conclusions. Moreover, the title mentions "AQP2 Membrane Targeting," but the study identifies the involvement of other AQPs as well. Further clarification is needed to determine whether these effects are direct or indirect consequences of the treatment.

We acknowledge the reviewers' concerns regarding the observed variability in the experimental data regarding water intake and urine output, and as a response, we have moderated our overall conclusions. It is widely recognized that studies utilizing metabolic cages can induce stress in mice, potentially leading to variability in measurements as we also observe. Due to ethical considerations regarding animal welfare, we were unable to allow for acclimatization of the mice to the metabolic cages prior to urine output measurements, thus introducing a potential bias.

Nevertheless, in an effort to transparently represent changes in urine output, we have provided individual data for each mouse, allowing for tracking of urine output over time (Figure 2A). Additionally, we have included measurements of urine osmolality, revealing a significant decrease following adenine diet compared to control mice. Furthermore, our data indicates a significant increase in urine osmolality during the recovery phase compared to adenine treatment alone, suggesting an enhancement in urine concentration mechanisms. In line with that, our data demonstrate a significant upregulation in the expression and apical membrane targeting of AQP2 during the recovery phase. Moreover, there was a significant increase observed in AQP3 expression post-recovery. Collectively, these findings strongly indicate that both AQP2 and AQP3 might play a role in the recovery of water homeostasis.

In light of our investigation, both direct and indirect effects of adenine could be involved in the regulation of the different AQPs. While the precise mechanisms of adenine's direct impact on AQPs regulation remain to be fully elucidated, it is plausible that adenine may exert direct effects via its interaction with Ade2R. Additionally, adenine's metabolite, 2,8-dehydroxyadenine (2,8-DHA), could play a role in AQP regulation, particularly in the context of 2,8-DHA nephropathy. Adenine metabolism may also influence cellular signaling pathways or osmotic balance, indirectly impacting AQPs expression and function. For example, alterations in intracellular osmolarity or cellular stress induced by adenine metabolism could trigger signaling cascades that modulate AQPs expression levels. We have provided further elaboration on this topic in the discussion section of our manuscript.

The comments are addressed in blue in the revised manuscript. 

References

  1. Klinkhammer, B.M., et al., Current kidney function parameters overestimate kidney tissue repair in reversible experimental kidney disease. Kidney Int, 2022. 102(2): p. 307-320.
  2. Klinkhammer, B.M., et al., Cellular and Molecular Mechanisms of Kidney Injury in 2,8-Dihydroxyadenine Nephropathy. J Am Soc Nephrol, 2020. 31(4): p. 799-816.
  3. Sabra, M.S., F.K. Hemida, and E.A.H. Allam, Adenine model of chronic renal failure in rats to determine whether MCC950, an NLRP3 inflammasome inhibitor, is a renopreventive. BMC Nephrol, 2023. 24(1): p. 377.
  4. Fong, D., et al., Renal cellular hypoxia in adenine-induced chronic kidney disease. Clin Exp Pharmacol Physiol, 2016. 43(10): p. 896-905.
  5. Guo, Y., et al., Huangjinsan ameliorates adenine-induced chronic kidney disease by regulating metabolic profiling. J Sep Sci, 2021. 44(24): p. 4384-4394.
  6. Ma, L.J. and A.B. Fogo, Model of robust induction of glomerulosclerosis in mice: importance of genetic background. Kidney Int, 2003. 64(1): p. 350-5.
  7. Ishola, D.A., Jr., et al., In mice, proteinuria and renal inflammatory responses to albumin overload are strain-dependent. Nephrol Dial Transplant, 2006. 21(3): p. 591-7.
  8. Gurley, S.B., et al., Impact of genetic background on nephropathy in diabetic mice. Am J Physiol Renal Physiol, 2006. 290(1): p. F214-22.
  9. Gurley, S.B., et al., Influence of genetic background on albuminuria and kidney injury in Ins2(+/C96Y) (Akita) mice. Am J Physiol Renal Physiol, 2010. 298(3): p. F788-95.
  10. Kishore, B.K., et al., Cellular localization of adenine receptors in the rat kidney and their functional significance in the inner medullary collecting duct. Am J Physiol Renal Physiol, 2013. 305(9): p. F1298-305.

Reviewer 3 Report

Comments and Suggestions for Authors

The author response reviewer's all comments with moderated modification of the manuscript. 

Author Response

We thank the reviewer and appreciate that there are no further comments. 

Round 3

Reviewer 1 Report

Comments and Suggestions for Authors

Thanks for the responses, I have no further comments.